Down-regulated NEDD4L facilitates tumor progression through activating Notch signaling in lung adenocarcinoma

Lin Liping 1 2
Wu Xuan 3
Jiang Yuanxue 1 2
Deng Caijiu 1 2
Luo Xi 1 2
Han Jianjun 1 2
Hu Jiazhu 1 2
Cao Xiaolong 1 2 caoxiaolong@pyhospital.com.cn
1 Department of Oncology, Panyu Central Hospital , Guangzhou, Guangdong Province , China
2 Cancer Institute of Panyu , Guangzhou, Guangdong Province , China
3 Department of Oncology, Peking University Shenzhen Hospital , Shenzhen, Guangdong Province , China
Gould Gwyn
Electronic publication date: 2022 May 24
Publication date: 2022
Volume: 10
Electronic Location ID: e13402
Received 2021 Dec 21; Accepted 2022 Apr 17
Copyright: © 2022 Lin et al.
Copyright year: 2022
Copyright holder: Lin et al.
License: This is an open access article distributed under the terms of the Creative Commons Attribution License, which permits unrestricted use, distribution, reproduction and adaptation in any medium and for any purpose provided that it is properly attributed. For attribution, the original author(s), title, publication source (PeerJ) and either DOI or URL of the article must be cited.
License URL: https://creativecommons.org/licenses/by/4.0/

Keywords: Lung adenocarcinoma, NEDD4L, Notch2, Hes1, RO4929097

Funding: Science and Technology Project of Guangzhou 202002030085 This work was supported by Science and Technology Project of Guangzhou (No. 202002030085). The funders had no role in study design, data collection and analysis, decision to publish, or preparation of the manuscript.

==============================
Neural precursor cell expressed developmentally down-regulated 4-like protein (NEDD4L), an E3 ubiquitin ligase, exerts an important role in diverse biological processes including development, tumorigenesis, and tumor progression. Although the role of NEDD4L in the pathogenesis of lung adenocarcinoma (LUAD) has been described, the mechanism by which NEDD4L promotes LUAD progression remains poorly understood. In the study, the correlation between NEDD4L level and clinical outcome in LUAD patients was analysed using the data from The Cancer Genome Atlas (TCGA) database. NEDD4L expression in LUAD cell lines and tissue samples was assessed through quantitative real-time PCR (qRT-PCR). The biological function of NEDD4L on regulating LUAD cell proliferation was tested with Cell Counting Kit-8 (CCK-8) assay in vitro, and mouse xenograft tumor model in vivo. We found that NEDD4L expression was significantly decreased in LUAD tissues and cell lines. Lower expression of NEDD4L exhibited a significantly poorer overall survival. Functionally, NEDD4L knockdown in H1299 cells accelerated cell growth, whereas NEDD4L overexpression in A549 cells repressed cell proliferation. NEDD4L overexpression also inhibited tumor xenograft growth in vivo. Mechanistically, NEDD4L decreased the protein stability of notch receptor 2 (Notch2) through facilitating its ubiquitination and degradation by ubiquitin-proteasome system. Consequently, NEDD4L negatively regulated Notch signaling activation in LUAD cells, and RO4929097 (a Notch inhibitor) treatment effectively repressed the effect of NEDD4L knockdown on LUAD cell proliferation. Taken together, these results demonstrate that down-regulated NEDD4L facilitates LUAD progression by activating Notch signaling, and NEDD4L may be a promising target to treat LUAD.

Introduction

Lung cancer is the primary reason of cancer-related death both in China (Cao & Chen, 2019) and worldwide (Sung et al., 2021). There are two main types of lung cancer, and approximately 85% of lung carcinoma are non-small cell lung cancer (NSCLC), in which lung adenocarcinoma (LUAD) is the most frequent histologic subtype (Huang et al., 2021; Balzer et al., 2018). While surgical excision can successfully treat early-stage LUAD patients, the majority of LUAD are diagnosed at advanced stage, with a 16% 5-year survival rate (Thawani et al., 2018; Shen et al., 2021). It is urgent to identify effective biomarkers to early diagnosis and treatment of LUAD.

NEDD4L is a member of NEDD4 family (Chen et al., 2021). Emerging studies have demonstrated that aberrant expression of NEDD4L is correlated with a wide variety of human diseases including kidney disease progression (Manning et al., 2021), interstitial pulmonary fibrosis (IPF) (Chen et al., 2021), renal fibrosis (Zeng et al., 2020), and ischemic brain damage (Kim, Chokkalla & Vemuganti, 2021). Chen et al. (2021) showed that NEDD4L represses IPF progression via enhancing ubiquitination-dependent degradation of β-catenin and suppressing hypoxia-inducible factor-1α signaling. NEDD4L inhibition results in a marked ischemic brain damage and aggravating post-ischemic motor dysfunction by regulating the protein stability of α-Synuclein (Kim, Chokkalla & Vemuganti, 2021).

NEDD4 and its homologue NEDD4L are two important members in NEDD4 family. Although both NEDD4 and NEDD4L are expressed in epithelial cells in the respiratory system, their biological roles in LUAD progression are different. Most studies demonstrated that NEDD4 functions as an oncogene by degrading tumor suppressor genes such as PTEN (Eide et al., 2013), Roundabout 1 (Bianchi et al., 2021), and Immunoglobulin and proline-rich receptor-1 (Amraei et al., 2020). Nevertheless, several studies reported NEDD4 as a tumor suppressor. Lu et al. (2016) demonstrated that NEDD4 represses intestinal tumor growth through inhibiting LEF1 and YY1. Recent studies identified the role of NEDD4L in repressing tumor progression in many types of tumours. Dong et al. (2021) reported that NEDD4L level is down-regulated in clear cell renal cell carcinoma (ccRCC) tissues, and forced expression of NEDD4L inhibits ccRCC cell proliferation and migration through degrading autophagy regulatory protein ULK1 (Dong et al., 2021; Zhao et al., 2021). Lower expression of NEDD4L predicts worse overall survival and disease-specific survival in renal clear cell carcinoma (Dong et al., 2021). Zhang et al. (2021) demonstrated that ALCAP2 (β, β-dimethyl-acryl-alkannin) treatment suppresses LUAD cell proliferation and invasion by increasing NEDD4L expression, resulting in a NEDD4L-dependent β-catenin degradation.

The Notch pathway is a highly conserved pathway that couples signalling from the membrane receptor to transcriptional regulation (Meisel, Porcheri & Mitsiadis, 2020). Notch pathway exerts a crucial role in embryonic development, cell proliferation, migration, and homeostasis (Edwards & Brennan, 2021). It has been reported that abnormal Notch signalling is correlated with a wide variety of human diseases such as tumor initiation and progression (Weijzen et al., 2002). In mammals, there are four Notch receptors (Notch1–4) that couple to multiple ligands (delta-like ligand 1–4, Jagged 1 and 2) to activate downstream target genes (hairy enhancer of split gens1–7, hey1–2, and Nrarp, etc.) expression (Edwards & Brennan, 2021). All four Notch receptors have been shown to be involved in cancer. Especially, the regulatory role of NEDD4L in Notch1 has been verified in breast cancer. Guarnieri et al. (2018) demonstrated that down-regulated NEDD4L accelerates Notch signalling activation through de-repression of Notch1. In the present study, the correlation of NEDD4L expression with clinical outcome in LUAD patients was explored, and the mechanism by which NEDD4L activates Notch signalling activation was investigated.

Materials and Methods

LUAD tissues and cell lines

Twenty-eight pairs of LUAD tissues and matched normal tissues (>2 cm from the outer tumour margin) were obtained from the Panyu Central Hospital after obtaining proper written informed consent from each donor. All clinical procedures were approved by the Ethics Committee of Panyu Central Hospital (Protocol code: PYRC-2021-087), and carried out in accordance with Declaration of Helsinki. LUAD cell lines (A549, H1299, and H1975) and a normal human bronchial epithelial cell line (16HBE) were obtained from Cell Bank of Chinese Academy of Sciences (Shanghai, China) and cultured in Dulbecco’s Modified Eagle’s Medium (DMEM, GIBCO BRL, Armonk, NY, USA) containing 10% foetal bovine serum (GIBCO BRL, Armonk, NY, USA) in humidified 5% CO2 environment at 37 °C.

qRT-PCR

Total RNA was isolated from LUAD cells and tissues with TRIzol (Sigma-Aldrich, St. Louis, MO, USA). First-strand cDNA was synthesized using total RNAs, M-MLV (Takara, Tokyo, Japan), and Oligo (dT) primers (Takara, Tokyo, Japan). The temperature protocol of reverse transcriptional PCR was 10 min at 70 °C, 2 min at 0 °C, and 60 min at 42 °C. qRT-PCR was carried out using SYBR Premix Ex Taq™ II (Takara, Tokyo, Japan) on 7500 qRT-PCR System (Thermo Fisher Scientific, Waltham, MA, USA). The temperature protocol for qRT-PCR was 20 min at 95 °C, followed by 35 cycles (15 s at 95 °C and 15 s at 60 °C). The sequence of NEDD4L primer pairs was AGAAAGGTCTTGACTATGGGGGT (sense) and AATTAGGGTTGATCTGAAGGGTGT (antisense). The sequence of β-actin (for internal control) primer pairs was AAATCTGGCACCACACCTTCTAC (sense) and AACATGATCTGGGTCATCTTCTCG (antisense). Relative mRNA level of NEDD4L was calculated with the 2(−ΔΔCT) method (Livak & Schmittgen, 2001).

Western blot

Total proteins were obtained from LUAD cells with RIPA buffer (Solarbio, Shanghai, China) and quantified by BCA protein quantification kit (Solarbio, Shanghai, China). Approximately 40 µg of total protein were separated through a 10% SDS-PAGE and were then transferred to PVDF membranes. Membranes were next incubated with the specific primary antibody against Notch1 (1:1,000, ab280898; Abcam, Cambridge, MA, USA), Notch2 (1:3,000, ab245325; Abcam, Cambridge, MA, USA), Notch3 (1:600, ab252845; Abcam, Cambridge, MA, USA), Notch4 (1:600, ab184742; Abcam, Cambridge, MA, USA), and ubiquitin (1:1,500, ab134953; Abcam, Cambridge, MA, USA) for 1 h at room temperature (RT). After washing with TBST five times, membranes were incubated with HRP-coupled anti-rabbit or mouse secondary antibody (1:6,000) for 1 h at RT. Immunoblots were visualized with the chemiluminescence detection system.

Co-immunoprecipitation (Co-IP)

A549 cells were treated with RIPA buffer (Solarbio, Shanghai, China) containing phenylmethylsulfonyl fluoride and incubated with NEDD4L antibody (1:200, ab240753; Abcam, Cambridge, MA, USA) for 14 h at 4 °C. Protein A/G magnetic beads (Thermo Fisher Scientific, Waltham, MA, USA) were added into IP mixture for 12 h at 4 °C. After washing with PBS, the precipitates were assessed through western blot analysis using Notch2 antibody (1:3,000, ab245325; Abcam, Cambridge, MA, USA).

Cycloheximide (Chx) chase assay

After NEDD4L overexpression, A549 cells were treated with Chx (80 µg/ml, Sigma-Aldrich, St. Louis, MO, USA) for the indicated times. Protein lysates were collected at 0, 2, 4, and 8 h exposure to CHX. The protein level of Notch2 was assessed using western blot analysis.

Overexpression and RNA interference (RNAi) of NEDD4L

The recombinant plasmids of pcDNA-NEDD4L which contain the full-length NEDD4L cDNA were constructed in our laboratory to overexpress NEDD4L. NEDD4L siRNA mixtures against NEDD4L (siNEDD4Ls) were obtained from GeneChem Co., Ltd. (Shanghai, China). Lipofectamine™ 3000 (Invitrogen, Waltham, MA, USA) was used to transfect pcDNA-NEDD4L or siNEDD4Ls into LUAD cells. The efficiency of overexpression and knockdown of NEDD4L has been validated in the study.

Cell proliferation assay

LUAD cell proliferation was assessed using CCK-8 reagent (Dojindo, Kumamoto, Japan). A549 cells were treated with pcDNA-NEDD4L to overexpress NEDD4L, and then NEDD4L-overexpressed A549 cells or A549 cells (2 × 103 cells in 96-well plates) were incubated with 10 µL CCK-8 at the indicated time point for another 1 h at 37 °C. H1299 cells were treated with siNEDD4Ls to inhibit NEDD4L, and then NEDD4L-knocked down H1299 cells or H1299 cells (2 × 103 cells in 96-well plates) were incubated with 10 µL CCK-8 at the indicated time point for another 1 h at 37 °C. The absorbance was assessed at 450 nm with a microplate reader (Thermo Fisher Scientific, Waltham, MA, USA).

Colony formation assay

NEDD4L-overexpressed A549 cells or A549 cells, and NEDD4L-knocked down H1299 cells or H1299 cells (approximately 300 cells each well) were seeded in 12-well plates. Fourteen days later, cells were fixed with 4% PFA and dyed using 0.1% crystal violet for 20 min, and cell colonies were calculated.

Xenograft mouse model

The animal experiment protocol was approved by the Ethics Committee of Panyu Central Hospital (PYRC-2021-086). All animal experiments were performed in accordance with the guidelines of Ethics Committee of Panyu Central Hospital to minimize the suffering of animals. Ten athymic BALB/c mice (approximately 6 week-old and 22 g) were obtained from the Shanghai Laboratory Animal Center, Chinese Academy of Science (Shanghai, China) and were maintained in a pathogen-free facility under 12 h light/dark cycle, at 25 ± 2 °C, with relative humidity 50% ± 10%. Mice were supplied with unlimited sterile chow and water. A549 cells stably expressing NEDD4L (A549/NEDD4L cells) were constructed. A549/NEDD4L cells and A549 cells (1 × 107 cells in 100 μL PBS) were subcutaneously inoculated into the right flank of BALB/c nude mice (n = 5), respectively. Tumour growth was surveyed at the indicated time-point. At the end of experiment, mice were euthanized under inhalational anaesthesia using 2% isoflurane, and xenografted tumours were harvested.

Statistical analysis

All data were represented as the mean ± standard deviation from three separate experiments. Statistical analysis was carried out through SPSS 18.0 (IBM Corp., Armonk, NY, USA), and two-tailed student’s t test was applied to compare differences between two groups. The difference was seemed as significant when p value <0.05.

Results

Down-regulated NEDD4L was associated with poor prognosis in LUAD

To explore the role of NEDD4L in LUAD, NEDD4L expression in LUAD was first analysed in TCGA database. As displayed in Fig. 1A, NEDD4L expression was significantly decreased in tumour tissues compared with normal controls. Moreover, decreased expression of NEDD4L was correlated with clinical stage (Fig. 1B). Furthermore, Fig. 1C showed that lower expression of NEDD4L exhibited a significantly poorer overall survival. Then NEDD4L expression was assessed in LUAD tissues and matched normal tissues. As shown in Fig. 1D, NEDD4L expression was decreased in most LUAD tissues. NEDD4L expression was also assayed in LUAD cell lines (A549, H1975, and H1299) and a human bronchial epithelial cell line 16HBE. Figure 1E showed that NEDD4L expression was significantly decreased in these LUAD cell lines compared with 16HBE cells. The frequent down-regulation of NEDD4L in LUAD implies that NEDD4L might function as a tumour suppressor in LUAD.

Figure 1 Downregulated NEDD4L was correlated with poor prognosis in LUAD.

(A) The NEDD4L expression in LUAD from TCGA (TCGA-LUAD) database was analysed through GEPIA platform (http://gepia.cancer-pku.cn/index.html). LUAD samples (n = 483) were showed as red, and normal samples (n = 347) were showed as grey. (B) The NEDD4L expression in TCGA-LUAD database was analysed in accordance with clinical stage through GEPIA platform. (C) The overall survival (OS) was analysed in TCGA-LUAD database in accordance with NEDD4L level. (D) LUAD tissues and matched normal tissues (n = 28) were used to assess the NEDD4L mRNA level using qRT-PCR. (E) qRT-PCR analysis of NEDD4L mRNA level in LUAD cell lines and 16HBE cells. *p < 0.05, **p < 0.01.

NEDD4L negatively regulated LUAD cell proliferation in vitro and in vivo

Then the biological role of NEDD4L in regulating LUAD cell proliferation was examined using CCK-8 and colony formation assay in vitro, and mouse xenograft tumor model in vivo. Given that the endogenous NEDD4L expression was the lowest in A549 cells and the highest in H1299 cells, NEDD4L was overexpressed in A549 cells (Figs. S1A and S1B) and inhibited in H1299 cells (Figs. S1C and S1D), and then cell proliferation was assessed. The results from CCK-8 assay revealed that forced expression of NEDD4L repressed A549 cell proliferation (Fig. 2A). Colony formation assay showed that NEDD4L overexpression reduced A549 cell colony formation ability (Figs. 2B and 2C). On the contrary, knockdown of NEDD4L accelerated H1299 cell proliferation (Fig. 2D) and increased H1299 cell colony formation ability (Figs. 2E and 2F). Furthermore, the role of NEDD4L in tumour growth in vivo was examined. As shown in Figs. 2G and 2H, overexpression of NEDD4L in A549 cells markedly repressed tumour growth of LUAD xenografts in nude mice.

Figure 2 NEDD4L negatively regulated LUAD cell proliferation in vitro and in vivo.

(A) NEDD4L was overexpressed in A549 cells, and then cell proliferation was assessed through CCK-8. (B and C) NEDD4L was overexpressed in A549 cells, and then cell growth was assessed through colony formation assay. NEDD4L was knocked down in H1299 cells, and then cell proliferation (D) and growth (E and F) was assessed using CCK-8 assay and colony formation assay, respectively. (G and H) NEDD4L-overespressed A549 cells (or A549 control cells, 1 × 107 cells in 100 μL PBS) were inoculated subcutaneously into BALB/c nude mice (n = 5). Tumour growth was surveyed at the indicated time-point. **p < 0.01.

NEDD4L promoted ubiquitination and proteasomal degradation of Notch2

To investigate the regulatory role of NEDD4L in Notch signalling, the Notch1–4 expression was measured in A549 cells after NEDD4L overexpression. Although NEDD4L did not change the mRNA expression of the four Notch receptors (Fig. S2A), NEDD4L overexpression significantly decreased the protein level of Notch1 and Notch2 (Figs. 3A and 3B), indicating that NEDD4L might promote Notch1 and Notch2 protein degradation as an E3 ubiquitin ligase. The results from immunofluorescence analysis showed that NEDD4L predominately located in cytoplasm in A549 cells (Fig. S2B), and Notch2 mainly located in cytoplasm and partly located in nucleus (Fig. S2C). Here the role of NEDD4L in regulating Notch2 protein stability was assessed because Notch2 protein decreased more than Notch1 protein in NEDD4L-overexpressed A549 cells. The interaction of NEDD4L with Notch2 was first assessed via Co-IP. Figure 3C showed that NEDD4L directly interacted with Notch2 in A549 cells. The results from CHX chase assay revealed that NEDD4L overexpression notably reduced Notch2 protein stability in A549 cells after treatment with CHX to block new protein synthesis (Figs. 3D and 3E). Notch2 ubiquitination was next assessed by Notch2 immuno-precipitation and subsequent western blot assay for ubiquitin. Figure 3F showed that forced expression of NEDD4L markedly increased Notch2 ubiquitination in A549 cells. Moreover, NEDD4L-overexpressed A549 cells were treated with a proteasome inhibitor (MG132), and the role of MG132 in inhibiting NEDD4L-mediated Notch2 degradation was assessed using western blot analysis. As shown in Fig. 3G, decreased expression of Notch2 was restrained after MG132 treatment. These results suggest that NEDD4L accelerates ubiquitination and proteasomal degradation of Notch2.

Figure 3 NEDD4L promoted ubiquitination and proteasomal degradation of Notch2.

(A) NEDD4L was overexpressed in A549 cells, and then cell proliferation was assessed through CCK-8. (B and C) NEDD4L was overexpressed in A549 cells, and then cell growth was assessed through colony formation assay. NEDD4L was knocked down in H1299 cells, and then cell proliferation (D) and growth (E and F) was assessed using CCK-8 assay and colony formation assay, respectively. (G and H) NEDD4L-overespressed A549 cells (or A549 control cells, 1 × 107 cells in 100 μL PBS) were inoculatedsubcutaneously into BALB/c nude mice (n = 5). Tumour growth was surveyed at the indicated time-point.

Down-regulated NEDD4L contributed LUAD cell proliferation by activating Notch signaling

We next investigated whether NEDD4L regulated Notch signaling activation through assessing their downstream target genes (NICD, Hey1 and Hes1). The results from qRT-PCR analysis showed that the expression of Notch2, NICD, Hes1, and Hey1 in murine tumor tissues from NEDD4L-overexpressed group was significantly decreased compared with control group (Fig. S2D). NEDD4L overexpression also repressed the mRNA and protein level of NICD, Hey1 and Hes1 in A549 cells (Figs. 4A and 4B). Contrarily, knockdown of NEDD4L enhanced the expression level of NICD, Hey1 and Hes1 in H1299 cells (Figs. 4C and 4D), indicating that NEDD4L negatively regulated Notch signaling activation. Finally, we demonstrated that NEDD4L knockdown accelerated LUAD cell proliferation, whereas pharmacological inhibition of Notch signaling by its specific inhibitor (RO4929097) significantly blocked the effect (Figs. 5A and 5B). Taken together, these data demonstrate that down-regulated NEDD4L facilitates LUAD progression through de-repression of Notch2 and thus activating Notch signaling.

Figure 4 NEDD4L negatively regulated the activation of Notch signaling.

(A and B) NEDD4L was overexpressed in A549 cells, and then the downstream target genes of Notch signaling (NICD, Hey1 and Hes1) were assessed using qRT-PCR (A) and western blot (B) analysis. (C and D) NEDD4L was knocked down in H1299 cells, and then the expression of NICD, Hey1 and Hes1 was assessed using qRT-PCR (C) and western blot (D) analysis. **p < 0.01.

Figure 5 Downregulated NEDD4L contributed LUAD cell proliferation by activating Notch signaling.

(A and B) NEDD4L was knocked down in H1299 and H1975 cells, respectively, and then cell proliferation was assessed using CCK-8 assay in the presence or absence of RO4929097 (25 μM). **p < 0.01.

Discussion

As a part of the post-translational protein modification, ubiquitin-proteasome system (UPS) acts as an important role in the vast majority of physiological and pathological processes including embryogenesis (Tsukamoto & Tatsumi, 2018), cell growth (Wang et al., 2021), DNA damage and repair (Daulny & Tansey, 2009; Wang et al., 2020), and tumorigenesis (Suber et al., 2018; Snoek et al., 2013). NEDD4L, an E3 ubiquitin ligase, has recently been identified as a tumour suppressor in many types of human cancers. However, little is known about the role NEDD4L in LUAD, and revealing the mechanism by which NEDD4L accelerates LUAD progression is essential. In the present study we demonstrated that: (i) NEDD4L down-regulation is correlated with poor prognosis in LUAD; (ii) NEDD4L represses LUAD cell proliferation in vitro and in vivo; (iii) NEDD4L facilitates ubiquitination and proteasomal degradation of Notch2; (iv) inhibition of NEDD4L accelerates LUAD cell proliferation by activating Notch signalling. These results define the important role of NEDD4L/Notch axis in LUAD progression, providing a promising target to treat LUAD.

NEDD4 (also known as NEDD4-1) and NEDD4L (also known as NEDD4-2) are the member of HECT E3 ubiquitin ligase family (He et al., 2020). NEDD4 expression is frequently increased in tumor tissues from LUAD patients (Amodio et al., 2010; Song et al., 2018). Overexpression of NEDD4 is associated with TNM stage and tumor metastasis (Song et al., 2018). NEDD4 overexpression accelerates lung cancer cell proliferation, migration, and drug resistance. Several target proteins of NEDD4 have also been identified, such as phosphate and tension homology deleted on chromosome 10 (Wang et al., 2007), leucine Rich Repeat Containing G Protein-Coupled Receptor 5 (Novellasdemunt et al., 2020), and phosphatidylinositol 4-phosphate 5-kinase α (Tran et al., 2018). Additionally, high NEDD4 expression predicts a poor prognosis in different types of tumors including LUAD (Shen et al., 2020), hepatocellular carcinoma (Hang et al., 2016), gastric cancer (Yang et al., 2012), and colorectal cancer (Eide et al., 2013).

Unlike NEDD4, NEDD4L exhibits more complex roles in tumor progression. In prostate cancer and gastric cancer, NEDD4L promotes cancer cell proliferation and is positively associated with tumor progression (Jiang et al., 2019; Hu et al., 2009). On the contrary, in patients with NSCLC, lower expression of NEDD4L exhibits a significantly poorer prognosis (Sakashita et al., 2013). Down-regulated NEDD4L expression is negatively associated with histological grade, lymph node metastasis, and pathological stage in NSCLC (Sakashita et al., 2013). Nevertheless, the effect of NEDD4L on LUAD remains poorly understood. In the present study, to define the role of NEDD4L in LUAD, the NEDD4L expression in LUAD was analysed in TCGA database and validated in tumour tissues. We found that NEDD4L expression was down-regulated in LUAD tissues compared with normal controls. Furthermore, lower expression of NEDD4L predicted a significantly poorer overall survival. In the following study, the reason for NEDD4L repression (genetic alterations or epigenetic modification) in tumour tissues will been further investigated. Functionally, overexpression of NEDD4L repressed LUAD cell proliferation and colony formation ability, whereas NEDD4L inhibition facilitated LUAD cell proliferation and increased LUAD cell colony formation ability. Especially, overexpression of NEDD4L in LUAD cells markedly inhibited tumour growth of LUAD xenografts in nude mice, indicating that NEDD4L functions as a tumour suppressor in LUAD.

In a previous study, the regulatory role of NEDD4L in Notch signalling has been revealed (Guarnieri et al., 2018). Guarnieri et al. (2018) demonstrated that NEDD4L expression is correlated with a better relapse-free prognosis in patients with breast cancer, and NEDD4L negatively regulates Notch signalling through repressing Notch1. To further investigate the correlation of NEDD4L with Notch signalling in LUAD, the expression of the four Notch receptors was assessed in LUAD cells after NEDD4L overexpression. The current data showed that NEDD4L repressed the protein level of Notch1 and Notch2. Furthermore, Notch2 protein decreased more than Notch1 protein in LUAD cells after NEDD4L overexpression, and NEDD4L accelerated Notch2 degradation through ubiquitin-proteasome system. Consequently, NEDD4L repressed Notch signaling activation, and pharmacological inhibition of Notch signaling by RO4929097 repressed cell proliferation in NEDD4L-decreased LUAD cells. In fact, NEDD4L functions as a tumor suppressor by targeting multiple oncogenic molecules in LUAD besides Notch2 (Chen et al., 2021). These results suggest that drugs which increase but not decrease NEDD4L expression might be beneficial for LUAD treatment.

Supplemental Information

Supplemental Information 1 Figure S1 raw data.

Click here for additional data file.

Supplemental Information 2 Figure S2 raw data.

Click here for additional data file.

Supplemental Information 3 Overexpression and knockdown of NEDD4L.

Recombinant plasmids (pcDNA-NEDD4L) were transfected into A549 cells, and then the mRNA (A) and protein (B) levels of NEDD4L were assessed using qRT-PCR and western blot analysis. siRNAs against NEDD4L (siNEDD4Ls) were transfected into H1299 cells, and then the mRNA (C) and protein (D) levels of NEDD4L were assessed using qRT-PCR and western blot analysis.

Click here for additional data file.

Supplemental Information 4 The association of NEDD4L with Notch pathway.

(A) qRT-PCR was carried out to assess NOTCH1-4 mRNA level in A549 cells after NEDD4L overexpression. Immunofluorescence analysis was carried out to assess the cellular localization of NEDD4L (B) and NOTCH2 (C) in A549 cells. (D) The mRNA levels of NOTCH2, NICH, Hes1, and Hey1 was assessed using qRT-PCR analysis in A549 cells after NEDD4L overexpression. **p < 0.01.

Click here for additional data file.

Supplemental Information 5 WB raw data.

Click here for additional data file.

Supplemental Information 6 Figure 1 raw data.

Click here for additional data file.

Supplemental Information 7 Figure 2 raw data.

Click here for additional data file.

Supplemental Information 8 Figure 3 raw data.

Click here for additional data file.

Supplemental Information 9 Figure 4 raw data.

Click here for additional data file.

Supplemental Information 10 Figure 5 raw data.

Click here for additional data file.

Supplemental Information 11 Author checklist.

Click here for additional data file.

Additional Information and Declarations

Competing Interests

Author Contributions

Animal Ethics

Data Availability

The authors declare that they have no competing interests.

Liping Lin conceived and designed the experiments, performed the experiments, analyzed the data, authored or reviewed drafts of the paper, and approved the final draft.

Xuan Wu conceived and designed the experiments, performed the experiments, analyzed the data, authored or reviewed drafts of the paper, and approved the final draft.

Yuanxue Jiang performed the experiments, prepared figures and/or tables, authored or reviewed drafts of the paper, and approved the final draft.

Caijiu Deng performed the experiments, prepared figures and/or tables, authored or reviewed drafts of the paper, and approved the final draft.

Xi Luo performed the experiments, analyzed the data, prepared figures and/or tables, and approved the final draft.

Jianjun Han performed the experiments, analyzed the data, prepared figures and/or tables, and approved the final draft.

Jiazhu Hu performed the experiments, prepared figures and/or tables, and approved the final draft.

Xiaolong Cao conceived and designed the experiments, performed the experiments, analyzed the data, prepared figures and/or tables, and approved the final draft.

The following information was supplied relating to ethical approvals (i.e., approving body and any reference numbers):

The animal experiment protocol was approved by the Ethics Committee of Panyu Central Hospital (PYRC-2021-086, PYRC-2021-087).

The following information was supplied regarding data availability:

The raw measurements are available in the Supplemental Files.

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
