# Peer review of "Down-regulated NEDD4L facilitates tumor progression through activating Notch signaling in lung adenocarcinoma"

_PeerJ, doi:10.7717/peerj.13402_

## Round 0.1 · original submission · Minor Revisions

As you will see below, three reviewers have provided a rather long list of points to be considered. I regard most of these to fall into the minor revision category, but emphasise that you should address all of the points raised.

Some further work is suggested by one of the reviewers (measurement of invasiveness, for example): I believe this would be informative, if the studies are available. Alternatively, the issue which is raised in that context could be more deeply discussed. I leave that decision to you.

Congratulations on a nice study. I look forward to seeing the revised version in due course.

Reviewer 1 ·

Basic reporting

The introduction provides some interesting findings about the role of NEDD4L in lung-related diseases, including lung cancer, but one actually misses background on the biochemical function of NEDD4L as a E3 Ubiquitin Protein Ligase. Also , the full "name" of NEDD4L (= NEDD4 Like E3 Ubiquitin Protein Ligase) is never explained. In addition, I miss some background related to the biochemical functions of NEDD4L, and its close relative, NEDD4, in particular their role in ubiquitination of proteins and degradation of target proteins by the proteasome. That should be the 1st paragraph of the introduction, and clearly relates to the findings related to NOTCH2 functions, explained later.

Then, one may also wonder about the relevance of NEDD4, in a study about NEDD4L: how closely related are these 2 family members? Do they partially overlap or complement each others functions; can one replace the others, etc. This would be relevant for understanding the effects of only knocking down one of these 2 related genes.

Smaller issues:

English language use is mostly okay, but there are a number of typos and glitches, even in the abstract, that need to be fixed. For example (from abstract, line 36): "NEDD4L facilitates LUAD progression by activating Notch signaling, and may be a promising target for treat LUAD." Should either mean "to treat" or "for treatment of"....

Line 89: "Total RNAs were isolated from LUAD cells"....should mean: total RNA was isolated

Line 117: "A549 cells were overexpressed with NEDD4L and were then treated with Chx". Thats just not a good sentence...

Experimental design

There are no major flaws with the description of the research methods. The authors have used a good spectrum of different methods that partly supplement each other, and covered mRNA and protein gene expression equally well. They have looked into effects on tumor and normal cells and even used mouse xenograft studies. In addition, when cell lines were used, typically more than just one line has been used and there should be no major discrepancies that arent covered here; such as when researchers only use a single cell line for all experiments.

It is also to mention that the authors did not only generate gene knock-down studies (by siRNA or shRNA vectors), but also tested the opposing concept: overexpression of the target gene, in comparison to down-regulation. Also here, different cell lines were used which complement each other. These data are rather trustworthy and the experimental design provides no points for major criticism.

One could question if the use of 10 nude mice for the xenograft studies provides sufficient statistical power, especially if there were minor phenotypic effects of NEDD4L-knockdown.

Otherwise, the statistical evaluation of results appears appropriate and there are no conclusions anywhere from doubtful statistical procedures or misinterpretations.

Validity of the findings

One may wonder why the cell lines have only partially reduced expression of NEDD4L since there appears such a positive benefit for loss of function of this target for cell proliferation and growth. Is it in fact gene dosage of NEDD4L expression and functions that are tightly regulated in these cells, while total loss of function appears to be avoided - also in tumor tissues? How do the authors explain this?

What is the reason for the repression of NEDD4L expression in tumor tissues - have the authors explored the possibility of genetic alterations (deletions) in at least some of the tumors? This could be at least discussed.

Generally, the reduction of NOTCH2 and particular NOTCH1 levels observed are not very pronounced, especially when significant changes of NODD4L levels are introduced. This raises the question if this is functionally really that significant - and if NEDD4L may rather have many pleiotropic effects on many different genes? This would be expected, as it is involved in regulating the function of the proteasome and protein degradation efficacy inside the cell. Would knocking down NOTCH2 expression (by siRNA?) result in comparable phenotypic effects? Pharmacological inhibition of NOTCH receptors is done (Fig. 4), but this targets all NOTCH receptors and downstream activities (plus a spectrum of off-target effects). Knocking down the NOTCH receptors specifically - in particular, NOTCH1 and 2 - would have been more conclusive.

The non-overlapping nature of NEDD4 versus NEDD4L is discussed in the last paragraph of the manuscript, and it is critical that this is done. This part, however, could also be at least mentioned in the introduction. It leaves the reader poised for the question the entire time you read the manuscript: are these 2 genes doing the same job?

Nevertheless, the question about biological redundancy remains: NOTCH1 and 2 may be affected - but how are expression levels of these 2 relative to EACH other? In figure 3, the effects of NEDD4L knock-down or overexpression on NOTCH1-4 proteins are shown. But Western blots are not really quantitative and cannot be compared to each other if different antibodies are used. What are the relative expression levels of NOTCH1-4 compared to each other? Is NOTCH2 maybe the most abundantly expressed of the 4 receptors? This should be addressed - it may further support the main findings and their interpretation. It appears that NOTCH2 is the only NOTCH receptor that is frequently affected by genetic mutations (usually amplifications) in lung adenocarcinoma. I would therefore almost expect that NOTCH2 is functionally more important in this cancer subtype than the other 3.

Additional comments

The main problem with this manuscript is probably related to the non-specificity of NEDD4L functions: it may target a large number of targets, of which NOTCH1 and 2 are just 2 examples. So this opens a bunch of questions that are actually less related to NEDD4L (and its relative NEDD4), but to NOTCH signaling. How do we know that an active NOTCH signaling pathway is a positive thing for the tumor cells? Only about 10% of the LUADs show overexpression of NOTCH receptors - and interestingly, this is mostly NOTCH2. IN almost all of these tumors, genetic amplification of the NOTCH2 locus can be observed. These tumors may indeed strongly benefit from increased NOTCH signaling - and reducing NOTCH levels would be expected to result in lower proliferation rates. However - what is with the rest of 80-90% of tumors that either shows no changes in NOTCH activity or expression - or even have inactivating mutations? It would be interesting to correlate expression levels and gain-of-function in NOTCH2 in the r tumor cell lines with the findings. Are these cell lines showing true activation of NOTCH2 activity, or NOTCH pathway activity; do they harbor amplification, or is NOTCH not really increased in these? That's, to me, an outstanding question.
If NOTCH activities in the 3 cell lines aren't fundamentally critical for tumor growth, and not highly activated in comparison to normal lung epithelia - then the NOTCH pathway may not be the best target for exploring NEDD4L functions.

·

Basic reporting

The manuscript is overall well-written. However, I would recommend the authors to proofread it once more due to some minor errors spotted during my read. To exemplify, a few of the errors are listed below:

1) Abstract: Line 2 "ubiquitin ligase" instead of "ubiquitinligase."
2) Line 120: "interference" in place of "interfere."
3) Line 136: "Colony formation assay" in place of "clone formation assay."

Experimental design

The experimental design has been sufficient to address the question of how NEDD4L impacts proliferation of lung adenocarcinoma cells. Within that scope, however, it does not provide evidence for decrease in the protein levels of Notch2 and/or its downstream molecules within the mouse tumors (for example: IHC of tumors with Notch2 Ab and NEDD4L).

Now, in terms of addressing the main question of how NEDD4L downregulation results in tumor progression via Notch2, the authors miss the point that tumor proliferation is only one facet of cancer advancement. Invasiveness, metastasis, stemness, etc., are other important features that define tumor progression.

It is highly recommend that the authors at least show whether impact on Notch2 protein levels by modulating NEDD4L levels causes any change in migratory and/or invasive tendencies of LUAD cells. (ref: ALCAP2 inhibits lung adenocarcinoma cell proliferation, migration and invasion via the ubiquitination of β-catenin by upregulating the E3 ligase NEDD4L; https://www.ncbi.nlm.nih.gov/pmc/articles/PMC8324825/).

The authors should also acknowledge in the 'Discussion' section the following:

1. NEDD4L modifies the protein levels of multiple other oncogenic molecules in lung adenocarcinoma and the effect on proliferation shown in the manuscript here to be via Notch2 downregulation is not the only mechanism. For example, PI3K/Akt pathway is also activated by Notch2 to increase proliferation of lung adenocarcinoma (ref: NEDD4L-induced ubiquitination mediating UBE2T degradation inhibits progression of lung adenocarcinoma via PI3K-AKT signaling; https://cancerci.biomedcentral.com/articles/10.1186/s12935-021-02341-9).

2. Since the authors have shown that NEDD4L expression is inversely correlated with stage of tumor, it is advisable to discuss the relevance of conducting future studies with different forms of mutated EGFR (commonly found in advanced LUAD patients) and checking whether upregulation of NEDD4L will be helpful in a certain subset of such patients (ref: Downregulation of NEDD4L by EGFR signaling promotes the development of lung adenocarcinoma; https://link.springer.com/article/10.1186/s12967-022-03247-4).

Validity of the findings

'no comment'

Additional comments

'no comment'

Reviewer 3 ·

Basic reporting

In this manuscript the authors compare the expression of NEDD4L between normal and tumor tissues and find that the protein is downregulated in tumor samples. Using various in vitro and in vivo assays they demonstrate that NEDD4L regulates proliferation and tumor progression via notch pathway.
I have several major and minor concerns that need to be addressed.
1. I would strongly recommend a immunohistochemical or immunocytochemical analysis of the involved proteins in cell lines or the primary tissues. As localization of proteins involved in notch pathway is critical for its location, the information is highly desirable.
2. Mention how far the normal tissue was from the tumor part.
3. What does 5/28 in line 169 signify?
4. Did the authors observe any correlation between levels of NEDD4l and morphology or proliferation rates in the cell lines?
5. The figures 3B and 2D need to be quantified.
6. In the results, briefly explain how to interpret the assay and results in Figure 3D.
7. In Figure 4, did they wrongly write NICD as NICH?
8. As PEERJ is an online journal, would it make sense to have color graphs which make it easier for the reader to distinguish between lines?
9. English language and grammar needs to be revised throughout the manuscript.

Experimental design

NA

Validity of the findings

NA

Additional comments

NA

---

## Round 0.2 · Minor Revisions

AS noted by the reviewer, there are still a number of typos, and formatting artefacts, that need to be fixed before this can be published.

There are also some occasions where words are not properly separated, possibly formatting between different programs.

I would urge you please spend some time addressing grammar and spelling throughout. Once that is attended to, I will accept the paper.

Reviewer 1 ·

Basic reporting

The authors have addressed most of the comments raised by the 3 independent reviewers - but not all. Some of the more critical issues have been properly addressed, such as adding critical points to the introduction, explaining the background of NEDD4L biology, and references have been added to track this previous research properly.

Some additional experimental work suggested by the reviewers has been accomplished, such as IF stainings, and added to the supplemental data.

I also understand that the authors did not further investigate some of the doubtless interesting aspects, such as an impact of NEDD4L over/underexpression on tumour cell invasiveness, motiltiy, stemness chaacteristics etc., as the current manuscript is already voluminous enough and sufficient.

However, other comments of the 3 reviewers have been simply discussed away. In some of these issues, I consider the responses as not sufficient but do not want to further meddle with the publication process.

This is understandable, and I believe the manuscript as such has still strongly improved after implementing the changes the authors decided to add.

Experimental design

There have not been too many comments by the 3 reviewers regarding the experimental procedures described, and the corresponding results. I personally think the results are fine, a large-enough spectrum of different methods has been used, and the results are described in the form of clear figures.

Validity of the findings

The observed effects on NOTCH1 and particularly 2 expressions are still not very high and may be borderline significant, which I consider the main flaw in this entire manuscript. But from our own experience, I can confirm that expression levels of these genes, and also downstream targets, often do not change dramatically while phenotypic effects are still observed.

Additional comments

there are still a number of typos, and formatting artifacts, that need to be fixed before this can be published. There are also some occasions where words are not properly separated, possibly formatting between different programs. But these things can be fixed in production.

Reviewer 3 ·

Basic reporting

The authors have addressed my concerns satisfactorily.

Experimental design

NA

Validity of the findings

NA

Additional comments

NA

---

## Round 0.3 · accepted · Accept

Thanks for your attention to these last points.